

# Unexpected characteristics of convective downdrafts in the upper-levels of tropical deep convective clouds

Sreehari Kizhuveettil[1], Jordi Vila-Guerau de Arellano[1], Martina Krämer[2,3], Armin Afchine[2], Luiz A. T. Machado[4], Martin Zöger[5], and Wiebke Frey[1]

[1]Meteorology and Air Quality group, Wageningen University and Research (WUR), P.O. Box 47 6700 AA, Wageningen, the Netherlands

[2]Institute of Climate and Energy Systems (ICE-4), Research Center Jülich, Jülich, Germany

[3]Institute for Atmospheric Physics (IPA), Johannes Gutenberg University Mainz, Mainz, Germany

[4]Instituto de Física, Universidade de São Paulo, São Paulo, Brazil

[5]Institute for Flight Experiments, German Aerospace Center, DLR, Oberpfaffenhofen, Germany

**Correspondence:** Sreehari Kizhuveettil (sreehari.kizhuveettil@wur.nl)

**Abstract.** This study investigates the thermodynamical and microphysical links of in-cloud downdrafts of tropical deep convective clouds using aircraft measurements from ACRIDICON-CHUVA field campaign focusing on the upper-levels (10 - 14 km). The Cloud Water Content (CWC) does not show a discernible trend with altitude or vertical velocity. This opposes the concept of condensate loading, enhancing the downdraft strength. Furthermore, the CWC in up- and downdrafts is found to be similar. The mean draft diameters exhibit a broadening trend with altitude in updrafts and downdrafts, while the mean air mass flux decreases with altitude. In the upper-levels, strong negative vertical velocities ($w < $ -2 m s$^{-1}$) are observed in the supersaturated region (RH$_{ice} > 110$ %), contradicting the general idea that downdrafts are driven/maintained by latent cooling. The spread in cloud particle number concentration was found to be similar in downdraft and updraft regions with a weak linear trend for $|w| > 1$ m s$^{-1}$ in transition ($90 \leq$ RH$_{ice} \leq 110$ %) and supersaturated regions. The mean particle size distributions (PSDs) indicate an increase in maximum particle size with altitude. Higher particle concentrations are observed in stronger drafts for particles with D$_p < 100$ m. Furthermore, the number concentration of larger particles (D$_p > 100$ m) increases faster in stronger drafts as altitude increases. Particle number concentrations in downdrafts are comparable to those in updrafts of similar strength at the same altitude. We speculate that large eddies that allow mixing between updrafts and downdrafts have an influence on the modulation of PSDs.



## 1 Introduction

Downdrafts play an important role in cloud structure as they are involved in complex dynamical, thermodynamic, and microphysical interactions across various scales. They are instrumental in modulating the cloud life cycle of precipitating convection; formation of precipitation in downdrafts cut off the convective updraft, thereby blocking the energy supply (Byers and Braham, 1948). Vertical motion in Deep Convective Clouds (DCCs) through updrafts and downdrafts couple the boundary layer with the free troposphere, facilitating the transport of mass, water, energy, aerosols, and pollutants from the boundary layer to the mid and upper troposphere (Cotton et al., 1995; Barth et al., 2015). They also bring free tropospheric properties to near-surface areas (Zipser, 1977; Betts, 1976), and in the upper cloud levels, can even enable transport from stratosphere to troposphere (Frey et al., 2014) and vice-versa. Consequently, the properties of updrafts and downdrafts have strong influence on the vertical transport of cloud condensate, cloud top height, and detrainment into anvils, which further influence the radiative balance (Del Genio et al., 2005).

Knupp and Cotton (1985) systematically discussed different types of downdrafts in convective clouds. According to them, fundamentally, downdrafts are facilitated by condensate loading and cooling by evaporation or melting. Precipitation-driven downdrafts which are located at the lower levels, are forced by precipitation loading, evaporation and melting. Downdrafts caused by cloud top and lateral entrainment of environmental air are termed as penetrative downdrafts. Further, their theoretical explanations based on earlier calculations, suggested that the overshooting downdrafts (downdrafts located in the overshoots of convective clouds) are caused by updrafts rising beyond the level of neutral buoyancy and quickly getting negatively buoyant which is distinctively different from lower level downdrafts. However, the earlier observations rarely probed above 7km altitude clouds, and thus, are not adequate for fully characterising the cloud structure.

In the past, a number of studies have discussed the effects of downdrafts in the lower troposphere. Downdrafts help in forming cold pools, which are regions of dense air formed by precipitation evaporation that propagate as density currents (Rotunno et al., 1988; Black, 1978) and could have devastating effects at the surface by means of strong winds. Coldpools are found to have a central role in shallow to deep convection transition (Khairoutdinov and Randall 2006) and are one of the factors controlling the deep convection in tropics (Tompkins 2001; Khairoutdinov and Randall 2006; Böing et al. 2012; Schlemmer and Hohenegger 2014; Kurowski et al. 2018). Fernández-González et al. (2016) showed that melting is crucial for maintaining downdrafts below the freezing level and preventing the collapse of low-level updrafts. Furthermore, their study found that mid-level downdrafts are influenced by the evaporative cooling of supercooled liquid water. Downdrafts also have implications for new particle formation in the outflow regions, which contribute to maintaining the boundary layer cloud condensation nuclei (CCN) concentrations in unpolluted environments (Wang et al., 2016; Williamson et al., 2019).

Downdrafts are typically expected to be formed at the mature stage of DCCs, as a consequence of precipitation formation. But downdrafts can form and precipitation can occur in the initial growing phase of a cloud. For example, cloud resolving simulations of intense deep convective clouds over Northern Australia (Hector storm system), reveal significant upper-level downdrafts associated with its overshooting clouds, even at developing stages (Frey et al., 2015). It is also argued that these rapid downdrafts associated with overshooting clouds can have important effects such as bringing the ozone rich stratospheric




air into troposphere. Additionally, Gerken et al. (2016) demonstrated the ability of convective scale downdrafts to transport free tropospheric ozone to the lower levels. In the event of strong downdrafts during thunderstorms with very high cloud top (greater than 10 km), an increase in surface ozone concentration has been observed with increasing cloud electrification (Unfer et al., 2025). These findings highlight the role of downdrafts in atmospheric composition and vertical transport processes, yet the impact of such downward mixing and transport remains largely uncertain.

In terms of mass flux, convective downdrafts have less but significant mass flux when compared to updrafts observed in both empirical studies (e.g., Yang et al. (2016b)) and Cloud Resolving Model (CRM) simulations (e.g., Mrowiec et al. (2012)).The mass flux of free-tropospheric air into the boundary layer has been proposed to play an important role for the equilibrium state of the tropical boundary layer (Raymond, 1995) with other processes such as entrainment at the boundary layer top.

The vertical motion and microphysics in the clouds have strong interaction through various processes such as droplet condensation-evaporation, and ice nucleation-sublimation. The particle size distribution (PSD) has an effect in modulating the downdraft intensity through modulating the cooling by evaporation. The abundance of small droplets can expose a greater surface area to evaporation since the surface-to-volume ratio is greater for small droplets compared to larger droplets for given liquid water content (Cotton et al., 2011). Studies of Yang et al. (2016a) using aircraft observations showed that liquid water content (LWC) and ice water content (IWC) are high in stronger updrafts in developing convective clouds. Grant et al. (2022) demonstrated a linear relationship between updraft vertical velocities within clouds and accumulated condensed water using high resolution simulations. However, there is no focus given for downdrafts and their microphysical structure.

Historically, in-situ measurements from aircrafts have provided information of the vertical motion in convective clouds (e.g., Byers and Braham (1948); Lenschow (1976); LeMone and Zipser (1980)). Byers and Braham (1948) provided a detailed explanation about dynamic and thermodynamic aspects of various stages of DCCs by analysing aircraft measurements. They oberved that, the updrafts extend to the upper troposphere, whereas the stronger downdrafts are primarily observed at lower tropospheric levels extending up to mid troposphere, enabled by evaporative cooling through the entrainment of environmental air. However, those earlier measurement campaigns had large uncertainties in their vertical velocity measurements because the aircrafts were not equipped with inertial navigation systems (LeMone and Zipser, 1980). Subsequent campaigns were able to eliminate these uncertainties and collected detailed statistics and vertical velocity structure of convective clouds in different locations worldwide (Houze Jr. and Betts, 1981; Jorgensen and LeMone, 1989; Lucas et al., 1994; Igau et al., 1999). Igau et al. (1999) analysed the tropical downdrafts from Tropical Ocean and Global Atmosphere Coupled Ocean Atmosphere Response Experiment (TOGA-COARE) and found that many tropical downdrafts are positively buoyant. Even after correcting for possible relative humidity excess, some of these downdraft cores stayed positively buoyant in their analysis. They argued that these downdrafts could be simply overshooting downdrafts or the downward-moving parts of gravity waves, or caused by more complex interplay of the forces generated by the surrounding three-dimensional convection and its immediate environment. Still, the measurements were heavily restricted to lower altitudes mainly due to aircraft limitations and thus, the upper level drafts are missing from these analyses.

Remote sensing methodologies are widely used to understand the convective cloud structure. Airborne and surface volumetric Doppler radars (e.g., Jorgensen and Smull, 1993; Sun et al. (1993)) and Radar Wind Profilers (RWP) (Giangrande et al.,



2013; Kumar et al., 2015; Schumacher et al., 2015) are some of the main sources to study the structure of vertical velocity
in convective clouds. These techniques enable simultaneous measurements at different heights (Tonttila et al., 2011) and thus,
can be used to probe a convective system throughout its passage over the measurement location. In a surface Doppler radar
based study, Sun et al. (1993) observed downdrafts in high altitudes (8 - 10 km) in the trailing and leading regions of a squall
line. Unlike their low-altitude counterparts, these downdrafts were positively buoyant with respect to the mean environment,
but were dynamically forced down to their equilibrium levels. It should be noted that the Doppler radar measures the hydrom-
eteor fall speed, and not the actual air velocity. Giangrande et al. (2016) analysed the radar wind profiler data set collected
during the 2 year Department of Energy Atmospheric Radiation Measurement Observations and Modeling of the Green Ocean
Amazon (GoAmazon2014/5) campaign and estimated convective cloud vertical velocity, area fraction, and mass flux profiles
for altitudes up to 12 km. Despite the advantages of remote sensing data, the necessary assumptions accounting for hydrome-
teor fall speed in observed Doppler velocity, which are used to estimate air velocity, make it less accurate than in-situ aircraft
measurements. Additionally, their inherent limitations related to beam width and attenuation in high altitude measurements
(above 12km) make it impossible to "see" the downdrafts at those altitudes. Therefore, in situ measurements remain essential
for accurately characterising the dynamics of convective clouds and in the developement of model parameterizations.

The "Aerosol, Cloud, Precipitation, and Radiation Interactions and Dynamics of Convective Cloud Systems - Cloud Pro-
cesses of the Main Precipitation Systems in Brazil" (ACRIDICON-CHUVA) campaign focused on tropical deep convective
clouds, and provides a unique opportunity to analyse dynamical, thermodynamic, and microphysical properties of downdrafts
in upper-levels (10 - 14 km). This study aims to analyse microphysical characteristics of downdrafts in tropical deep convective
clouds during the ACRIDICON-CHUVA campaign to understand the physical processes behind the observed characteristics,
focusing on the upper-level downdrafts which are not well studied in the past.

The study is organized as follows: Section 2 describes the instruments and dataset used in this study. Section 3 outlines the
methodologies employed for data processing. Section 4 first presents the draft statistics, then examines the observed micro-
physical characteristics of downdrafts. Finally, Section 5 concludes the study.

## 2 Dataset and Instruments

### 2.1 ACRIDICON-CHUVA field campaign

This study uses the ACRIDICON-CHUVA flight campaign data (Wendisch et al., 2016) conducted during the period of
September-October in 2014 over Amazonia, Brazil. The campaign aimed to investigate tropical deep convective clouds and
precipitation, focusing on various research topics such as cloud vertical evolution and life cycle, cloud processing of aerosol
particles and trace gases, and vertical transport and mixing, employing different flight strategies.

The campaign included 14 research flights with the German High Altitude and Long Range Research Aircraft (HALO) and
has collected nearly 13 hours of in-situ cloud data. The high endurance of the HALO aircraft enabled it to conduct extensive
flights, which carried out detailed and sophisticated measurements including in situ cloud particle observations across regions
with varying meteorological conditions. This include measurements from altitudes upto 15 km which is rare. The combination



of microphysical and dynamical measurements provide a unique opportunity to study the complex dynamical-microphysical interactions within the cloud. Here, the focus will be on measurements of vertical velocity (w), Cloud Water Content (CWC), cloud particle number concentration (N), and particle size distributions (PSDs). However, the biases in the sampling strategies and flight patterns affect the statistics and may not necessarily reflect the statistics as in nature, particularly at the core of the deep convection. The instruments from which the measurements are retrieved are briefly described below.

### 2.1.1 BAHAMAS

The vertical wind speed measurements and other meteorological and aircraft variables such as temperature (T), pressure (P), and True Air Speed (TAS) were obtained from the Basic Halo Measurement and Sensor System (BAHAMAS) (Krautstrunk and Giez, 2012) with 1 Hz temporal resolution. The 3-D wind measurements were calibrated following Mallaun et al. (2015), resulting in an uncertainty of 0.3 m/s for the horizontal components and 0.2 m/s for the vertical components. The absolute error value of temperature and static pressure is 0.1 K and 8 Pa respectively (Giez et al., 2023).

### 2.1.2 SHARC

The Relative Humidity with respect to ice ($RH_{ice}$) is retrieved from the gas phase mixing ratio of water ($H_2O_{gas}$) measured by the Sophisticated Hygrometer for Atmospheric ResearCh (SHARC) which is a tunable diode laser (TDL) hygrometer developed at DLR Flight Experiments. It is a closed-cell hygrometer which uses the absorption line of water vapor at 1.37 $\mu$m. The measurement range is from 10 to 50000 ppm and the overall uncertainty is 5 % relative and $\pm$ 1 ppm absolute uncertainty with 1Hz temporal resolution (Kaufmann et al., 2018).

### 2.1.3 NIXE-CAPS

The cloud particle sizes and number concentrations are retrieved using New Ice eXpEriment - Cloud and Aerosol Particle Spectrometer (NIXE-CAPS) and provided at a 1 Hz temporal resolution. There are two instruments incorporated in NIXE-CAPS: the NIXE-CAS-DPOL (Cloud and Aerosol Spectrometer with detector for polarization) and the NIXE-CIPg (Cloud Imaging Probe greyscale). The NIXE-CAS-DPOL instrument utilizes the forward scattered light by aerosol particles, water droplets, and ice crystals to determine particle size of the particles between 0.61 to 50 $\mu$m based on Mie Scattering theory of spherical particles. Additionally, it extracts particle asphericity from the polarization state of the backward scattered component of laser light. The NIXE-CIPg, an optical array probe, records two dimensional images of cloud droplets and ice particles in the size range between 15 $\mu$m and 937.5 $\mu$m with a resolution of 15 $\mu$m, using the maximum dimension. From the image, the particle shape as well as the particle size can be derived. On average, both NIXE-CAS-DPOL and NIXE-CIPg instruments exhibits an uncertainty of about 20 % in measuring the size of the particles and the number concentration (Meyer, 2012). Combining NIXE-CAS-DPOL and NIXE-CIPg outputs, particles with diameters between 0.61 and 937.5 $\mu$m can be sized and counted in 71 bins to obtain cloud particle size distributions.



## 3 Methods

This section outlines the methodologies employed for data processing in this study.

### 3.1 The "in-cloud" data

To analyse the dynamical and microphysical characteristics of deep convective clouds, it is essential to classify the data for cloudy points where measurements indicate the presence of clouds. This is achieved using Cloud Water Content (CWC) measurements calculated from PSDs. Cirrus and mixed phase CWC are calculated according to mass-Diameter relationship (as in Krämer et al. (2016)), whereas the liquid CWC is calculated directly from PSDs of cloud particles 3-937 $\mu$m. A non-zero value of CWC indicate the cloudy point. A more detailed explanation can be found in Krämer et al. (2020). This method

provides accurate flags for "in-cloud" data, which can later be used for filtering and identifying the cloudy drafts.

### 3.2 Defining updraft and downdraft

The updrafts and downdrafts represent the ascending and sinking motion in the atmosphere. Positive vertical velocity indicates updrafts, while negative vertical velocity indicates downdrafts. Although various methodologies have been utilized to identify the draft regions in previous studies (e.g, LeMone and Zipser (1980); Igau et al. (1999); Yang et al. (2016b)), they all funda-

mentally rely on vertical velocity measurements. In this study, we adopt the approach by LeMone and Zipser (1980), defining drafts as regions where the absolute value of vertical velocity exceeds 0 ms$^{-1}$ continuously for at least 500 meters. The measurement width is calculated from the TAS parameter by aggregating the TAS (distance covered per second) associated with observations based on the duration of a particular draft measurement in seconds. For all the analysis presented in this study, we use the above method combined with in-cloud criteria explained in section 3.1 to identify "cloudy" updrafts and downdrafts.

## 165 4 Results

### 4.1 Draft statistics

In order to get an understanding of the nature of updrafts and downdrafts observed during ACRIDICON-CHUVA, the draft statistics in terms of draft diameter and air mass flux associated with all the drafts detected from the dataset are analysed in this section. Cloud properties vary due to different factors such as altitude, life stage of the cloud system, etc. However, it is

difficult to characterize the life stage from the measurements used in this study.

The altitude-wise statistics are presented in Fig. 1. The Probability Density Function (PDF) of draft diameter in Fig. 1a represents the probability density values of drafts in certain diameter bin at a given altitude bin. Compared to upper-levels, less drafts are observed in lower levels. This is due to the measurement strategy that prioritised upper level measurements over the stratiform cloud deck. Generally, narrower drafts occur more frequently than the wider drafts. This is true for both updrafts

and downdrafts throughout the altitudes. Drafts at altitudes below 6 km mostly have a diameter less than 3000 m. Whereas the drafts with diameter greater than 4000 m are present only above 8 km and slightly more frequent above 10 km. This might



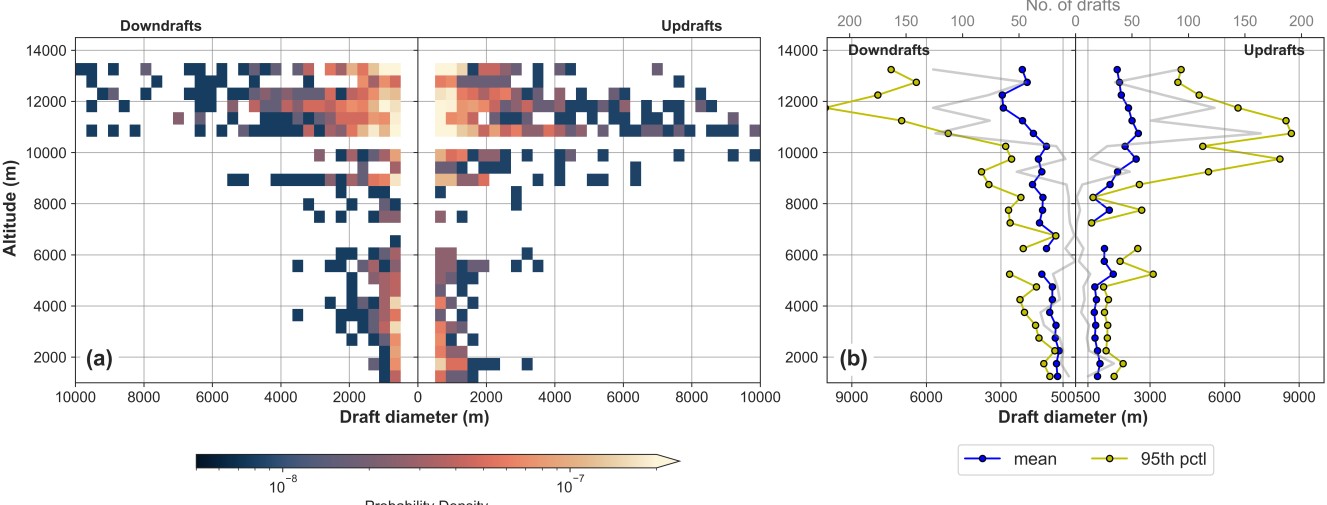

**Figure 1.** Altitude wise draft diameter statistics of all in-cloud drafts. (a) Probability Density Function (b) mean (blue) and $95^{th}$ percentile (yellow) values of diameter of drafs.

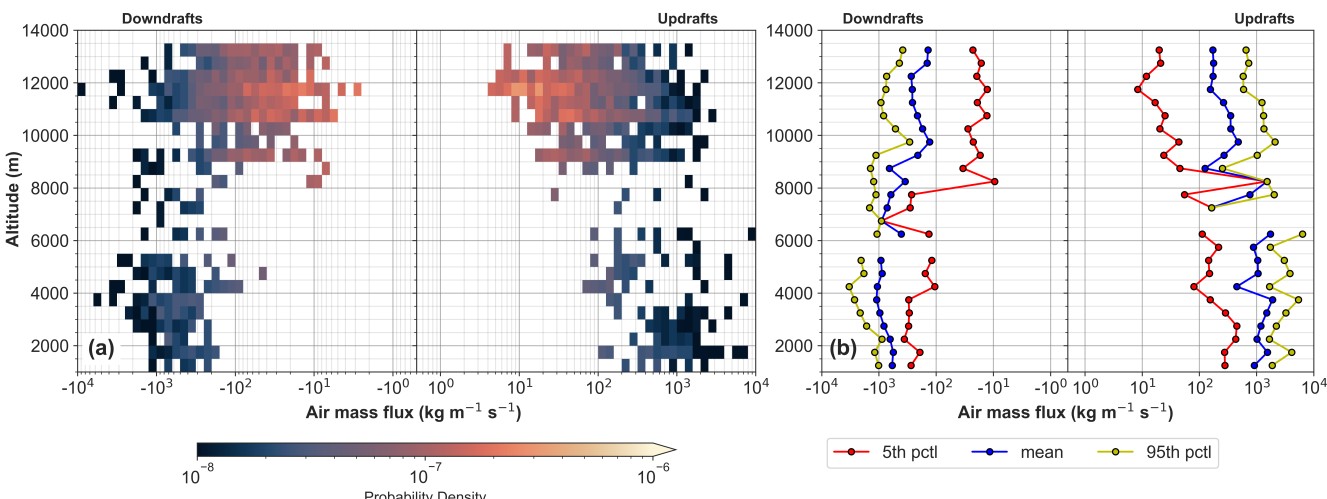

**Figure 2.** Altitude wise air mass flux statistics of all in-cloud drafts (a) Probability Density Function (b) mean (blue), $5^{th}$ percentile (red) and $95^{th}$ percentile (yellow) of air mass flux.

not be a characteristic vertical variation due to the significantly larger number of measurements of drafts at altitudes above 10 km. The vertical variation of mean and percentile values of draft diameter is presented in Fig. 1b. The mean value increases slightly in the lower altitudes (up to 6 km) and stays fairly constant up to 10 km. Above that, the mean draft width increases and

peaks at approximately 12 km and decreases. On the other hand, $95^{th}$ percentile increases with altitude. The peak is observed





around similar altitude as observed in the mean value curve. Much alike to the downdrafts, the mean value and $95^{th}$ percentile curves of the updrafts follow similar structure in the lower altitudes, after which it increases and peaks between 9.5 - 10 km. Both maximum $95^{th}$ percentile draft diameter and mean diameter values are higher in downdrafts (10079 m, 2955 m) than their updraft counterpart (8682 m, 2521 m). The larger draft diameters could be part of mesoscale updrafts and downdrafts in

the dissipating anvil clouds, i.e., when cloud top collapses (downdrafts), or in the clouds of the forming anvil (updrafts). The broadening of drafts with altitude observed in Fig. 1 is coherent with previous studies (e.g, Yang et al. (2016b); LeMone and Zipser (1980)) which, however, only extended up to 10 km. The data presented here, provide an extension to upper cloud levels (10-14 km) where the drafts are observed to broaden when compared to lower altitudes.

The air mass flux (M) is a measure of the amount of air transported by the draft. This can be calculated using the expression

M = $\overline{\rho}\,\overline{w}$ D, where $\overline{\rho}$ is the mean density of the draft, $\overline{w}$ is the mean vertical velocity, and D is the diameter of the draft. The updraft and downdraft air mass flux statistics are presented in Fig. 2. The PDF of air mass flux for updrafts and downdrafts (Fig. 2a) is almost identical throughout the altitudes. Drafts below 8km have minimum absolute air mass flux of 25 kg m$^{-1}$ s$^{-1}$ for updrafts and 48 kg m$^{-1}$ s$^{-1}$ for downdrafts. Whereas, drafts with weaker absolute air mass flux values ($< 10^1$ kg m$^{-1}$ s$^{-1}$) can be seen at altitudes above 8 km. This is due to the differences in vertical velocity structure at lower and higher altitudes and

exponentially decreasing air density profile. Majority of the drafts at higher altitudes show very weak mean vertical velocity. So, despite the fact that they are wider, the over all mass flux values are low. Drafts in lower altitudes correspond to less width but relatively high mean vertical velocity values, thus high mass flux. The mean and percentile statistics of air mass flux is shown in Fig. 2b. Both updraft and downdraft air mass fluxes show a decreasing trend with altitude. The percentile curves closely follow the mean curves with similar vertical variations.

It is noteworthy that the upper level drafts have smaller air mass flux values and wider draft diameters compared to the lower levels. Due to larger number of upper-level measurements, the updrafts and downdrafts observed between 10 and 14 km altitude alone cover almost 39 % and 44 %, respectively, of the total updrafts and downdrafts detected, which translates to 28 % of total updraft air mass flux and 31 % of total downdraft air mass flux in this campaign. The PDF for upper level (10-14 km) air mass flux versus draft diameter of updrafts and downdrafts is given in Fig. 3. Consistent with our previous analysis,

the updrafts and downdrafts in the upper level show similar characteristics and are almost mirrored in Fig. 3.

Overall, observed vertical velocities are relatively weaker in this campaign compared to previous studies (LeMone and Zipser, 1980; Yang et al., 2016b). However, the observed air mass flux is in the same order of magnitude. This could be due to that the observed drafts are slightly wider, and compensate for the weaker vertical velocities. The relationship between mean air mass flux and draft diameter shown in Fig. 3 for upper-level drafts are found to have a similar relationship as lower-level

drafts analysed in previous studies (e.g., Yang et al. (2016b)) with draft diameter increasing with air mass flux. The in-cloud mass flux presented here is only representative of convective-scale motions and may differ from mass flux at different scales, such as that of a Global Climate Model (GCM) grid. Giangrande et al. (2016) analysed the mass flux of convective clouds in Amazonia using radar wind profilers across larger spatial domains suitable for GCM-scale comparisons. Their findings show that mass flux at these scales is much smaller (approximately 100 times) than the values reported in this study. This is because

they used convective area fraction in the GCM-scale grid box, whereas current study utilizes the draft diameter.





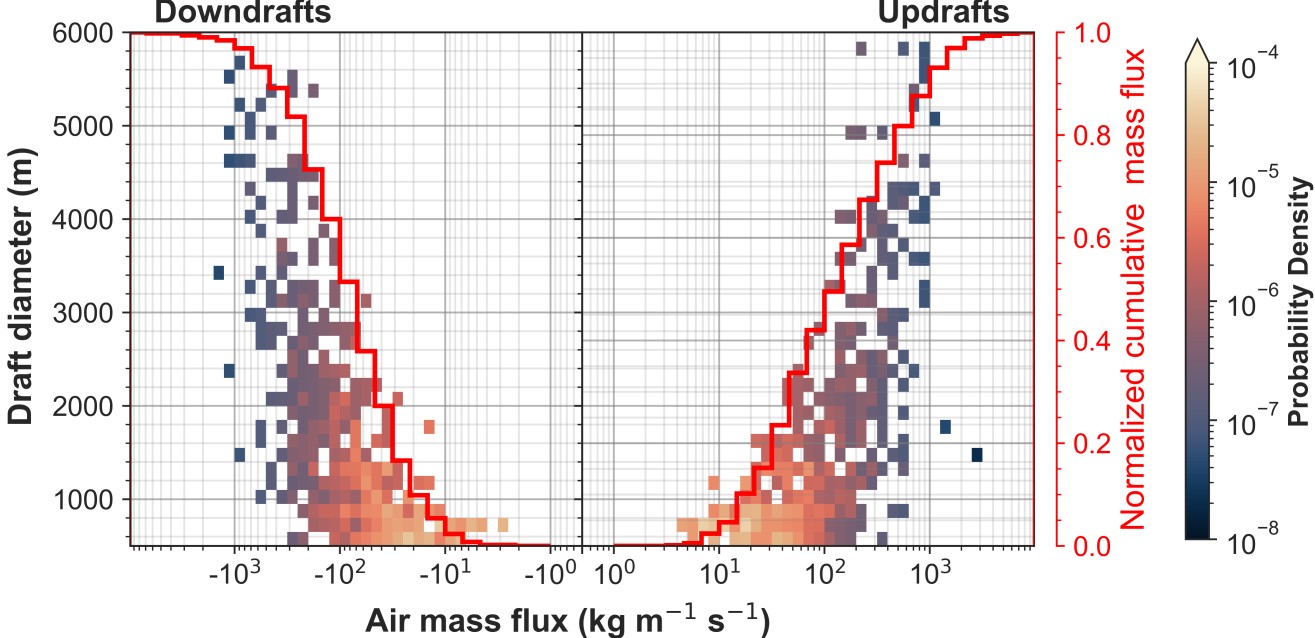

**Figure 3.** Probability Density of air mass flux vs draft diamter for in-cloud up and downdrafts with width greater than 500 m at altitudes 10 - 14km.

While we have defined drafts to have a width of at least 500m (Section 3.2), other studies used lower thresholds (e.g. Yang et al. (2016b)). In order to test the sensitivity of chosen draft width threshold, we repeat our analysis, including drafts between 100-500m (Fig. S1-S3). Due to the high aircraft speed of HALO, single observations will already be classified as larger than 100m drafts. Drafts with diameters 100-500 m contribute to more than 70 % of total up and down drafts detected, with almost

30 % contribution to updraft and downdraft air mass flux. The mean profiles show smaller values for diameter and air mass flux (Fig. S1b and S2b) with the inclusion of narrow drafts. The $95^{th}$ percentile and $5^{th}$ percentile have also decreased due to the presence of more lower values in the distribution in both draft diameter and air mass flux. Nevertheless, the mean draft diameter still shows a broadening (Fig. S1b). The inclusion of narrower drafts increased the frequency of weaker air mass flux values in all altitudes (Fig. S2). The air mass fluxes thus, show almost equal strength in the upper and lower levels. This result is

similar to Yang et al. (2016b), who found no obvious trend between air mass flux and altitude. Contrary to Yang et al. (2016b), the updrafts and downdrafts in the upper level are mirrored (Fig. S3). We would like to emphasize again that our study does not include the core of the convective system due to the aircraft limitations in probing regions with strong vertical motions, which is strongest in dry season (Giangrande et al., 2023; Machado et al., 2018).





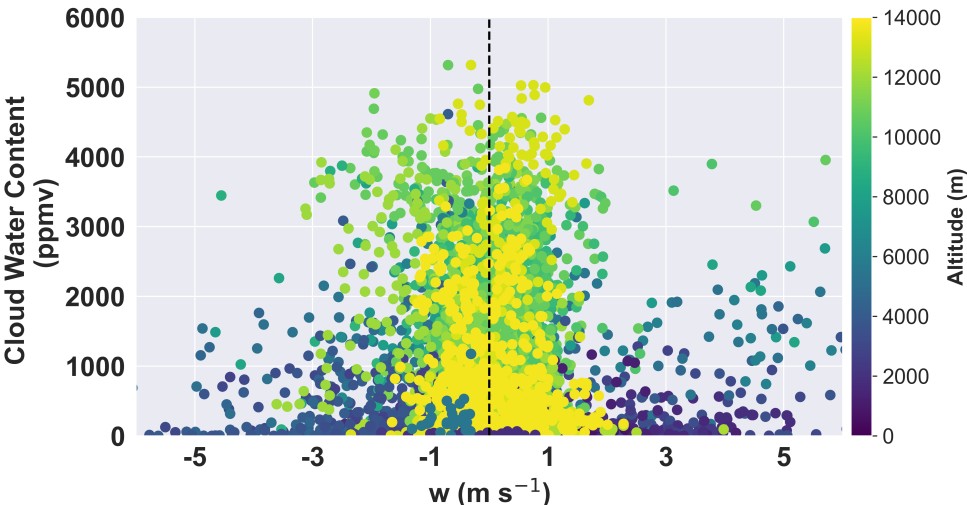

**Figure 4.** Cloud water content (ppmv) versus vertical veclocity (m/s) for all measurements. The colouring indicates the altitude of the observation.

## 4.2 Downdrafts and cloud microphysical properties - Cloud water content

In this section, we examine the observed characteristics of updrafts and downdrafts in terms of its bulk cloud microphysical properties. Figure 4 shows the Cloud Water Content (CWC) from all measurements that meet the draft criteria explained in section 3.2 as a function of vertical velocity to closely view the variations. The maximum observed CWC is below 6000 ppmv. From the figure, it can be noted that the structure and magnitude of CWC are qualitatively similar in both updraft and downdraft regions. Additionally, the high and low values are present in all altitudes and no particular effect of altitude is observed in CWC

characteristics.

The vertical motion in the atmosphere is regarded as strongly affected by the hydrometeors present in different regions. For example, the updrafts influence the supersaturation, thus the condensation process. Grant et al. (2022) demonstrated this effect by highlighting a robust linear relationship between the updraft vertical velocity and condensed water from a range of cloud resolving model simulations with the horizontal resolutions between 250-300 m of tropical and midlatitude deep

convective cloud systems. Additionally, the weight-induced drag can lead to downdrafts, which is one of the main drivers for precipitation-driven downdrafts (Houze Jr, 2014, p.190). Thus, higher CWC values should correspond to stronger downdrafts due to the higher drag force. However, this is not observed in our analysis even at the lower altitudes where the precipitation-driven downdrafts are located.

To be able to identify the effect of evaporative cooling on downdrafts in the upper-levels (10-14km, thus, rather cooling by

sublimation), we classify the data into three regimes based on $RH_{ice}$: a subsaturated regime ($RH_{ice} < 90\%$) , a transition/intermediate regime ($90\% < RH_{ice} < 110\%$) and a supersaturated regime ($RH_{ice} > 110\%$). The measurements of $RH_{ice}$ could be related to different parts of the cloud top (convective, outflow, cirrus etc.). For example, supersaturation with respect to ice is





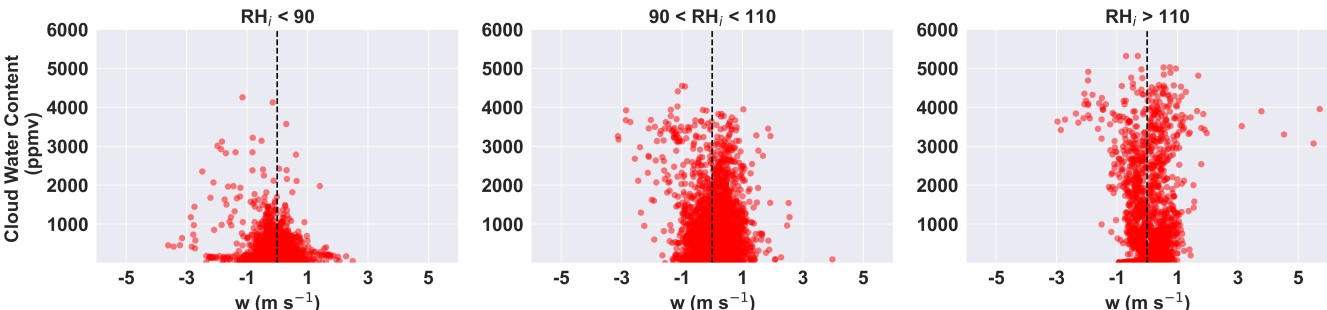

**Figure 5.** Cloud water content (ppmv) versus vertical veclocity (m/s) for upper-levels (10-14 km) in regimes based on $RH_{ice}$. (a) Subsaturated ($RH_{ice} < 90\%$), (b) Transition/Intermediate ($90\% < RH_{ice} < 110\%$), and (c) Supersaturated ($RH_{ice} > 110\%$).

very common in cirrus clouds (Comstock et al., 2004). A study based on lidar measurements in the HALO aircraft have shown that the $RH_{ice}$ distributions of cirrus clouds vary at different stages of the cloud's lifetime where $RH_{ice}$ modes are close to
saturation in young clouds, supersaturated in mature clouds and subsaturated in dissipating clouds (Dekoutsidis et al., 2023). Since we are classifying all the avaialble measurements based on $RH_{ice}$ and not solely focused on a particular cloud system, the presented results do not help directly distinguish the life stage of cloud from $RH_{ice}$ alone.

In the results shown in Fig. 5, the subsaturated regime generally contains smaller CWC values, as ice particles sublimate here, whereas the supersaturated regime contains the highest CWC values. The 99th percentile values of CWC in subsaturated and
supersaturated regimes are 1586 ppmv and 4442 ppmv respectively. We observe strong downdrafts in the supersaturated regime where, due to the supersaturation, sublimation is not possible. This contradicts the generally accepted view of evaporatively driven downdrafts, or sublimative driven here (10-14km), which should even be stronger, as sublimation requires more heat than evaporation. Gierens and Brinkop (2012) have shown that updrafts maintain ice supersaturation as they bring in moisture from the lower layers, however, in Fig. 5, updrafts are found in subsaturated and transition regimes. On the contrary, downdrafts
are not expected to bring in moisture to maintain supersaturation as they are coming from the upper layers. Note that the aircraft measurements do not provide the information on the evolution of a downdraft which can later change its saturation state due to lateral mixing with the surrounding updrafts or environment outside the cloud. Additionally, sublimation at an earlier time might have caused the downdraft to be saturated at the time of measurement.

While evaporation of raindrops is not sufficient to maintain saturation in precipitation-driven downdrafts (Kamburova and
Ludlam, 1968), convective-scale saturated or near-saturated downdrafts are observed in organized convective systems close to the convective core and at low levels (typically below 700-800 hPa). This is likely due to the cloud water supplied by convective updrafts, which helps maintain their saturated or near-saturated state (Zipser, 1977). Additionally, the latter study suggests that the mesoscale downdrafts developing behind the squall line system are typically unsaturated and driven by evaporative cooling. Since the upper-level measurements of this study are from cloud deck and also contain mesoscale downdrafts, this
could indicate that additional processes are at play to drive these downdrafts even in supersaturated regime. Thus the presence





**Figure 6.** Cloud particle number concentration (3 - 937$\mu$m) for upper level (10-14 km) in-cloud drafts (a) scatter plot and (b) for observations with $|w| > 1$ m/s as boxplot, in vertical velocity bins. The colours indicate different RH$_{ice}$ regimes denoted in the legend. The coloured lines represents the corresponding linear regression lines.

of downdrafts in supersaturated regime are an interesting feature to explore further with numerical simulations as they provide potential to also focus on the evolution of drafts.

The results shown in Fig. 4 and Fig. 5 are obtained from analysing all the upperlevel draft measurements including the weaker ones which are likely part of mesoscale drafts. By eliminating the observations with weaker velocities ($|w| < 1$ m s$^{-1}$),
we can more closely analyse the effect of saturation on bulk microphysics and vertical velocity of higher intensity drafts. The





measurements of number concentration for cloud particle sizes between 3 - 937 $\mu$m is utilized as it gives a better perspective on cloud particle processes in the drafts. The spread of cloud particle number concentration with vertical velocity values in updrafts and downdrafts is large in all $RH_{ice}$ regimes (Fig. 6a). Additionally, measurements in subsaturated regime with CWC value below 300 ppmv create an alternate branch in downdrafts which is analysed seperately. This is not visible in updrafts.

In Fig. 6b, drafts are categorized based on $RH_{ice}$ and binned in 0.5 m s$^{-1}$ vertical velocity bins to get a quantitative sense of the relationship. The extreme bins include all the measurements beyond $\mp$ 2.5 m s$^{-1}$. Overall, these show weak linear relationships (low r$^2$ values, refer Table A1&A2) in all the regimes and relatively higher slopes in downdrafts in the intermediate/transition and supersaturated regimes. The increasing trend in cloud particle number concentration with downdraft intensity suggests processes that produce more particles. For instance, ice multiplication processes have significant impact on number

concentrations in Numerical Weather Prediction (NWP) model simulations and idealized Large Eddy Simulations (LES) (Han et al., 2024). Although their dynamical link is still not understood, it is well conceivable that in more intense drafts, more turbulence might provide the possibility of ice particles colliding, increasing chances for breakup or splintering processes, leading to larger ice particle number concentrations.

### 4.3    Particle size distributions in downdrafts

In order to analyse the interaction of downdrafts with cloud microphysics, we use the PSDs from NIXE-CAPS. The size distributions are categorized in altitude bins 1.0 - 2.5 km, 2.5 - 5.0 km, 5.0 - 7.5 km, 7.5 - 10.0 km, and 10.0 - 14.0 km. The PSD in each altitude bin are further split according to absolute values of vertical velocity: 0 to 1 ms$^{-1}$, 1 to 3 ms$^{-1}$, and the last bin contains all the observations having absolute vertical velocity greater than 3 ms$^{-1}$. This is done for updrafts and downdrafts separately. The mean size distributions for each category are shown in Fig. 7 which exhibits changes with respect

to the altitude and intensity of vertical velocity. The freezing level (273.15 K) is located around 4.5 km during the campign. From the observed temperature range (as indicated in the figure panels), it can be deduced that, clouds in the altitude above freezing level up to 10 km are in mixed-phase regime (Jäkel et al., 2017; Cecchini et al., 2017b) with temperatures between 273.15 K to 235 K. The temperatures in altitudes 10 - 14 km are below homogenous freezing level (235 K). One can note that the highest updrafts and downdrafts ($|w| > 3$ ms$^{-1}$) show the largest number concentrations for particles with size $D_p <$

100 $\mu$m in altitude ranges above 5 km. The separation between mean PSDs for particles $D_p < 100$ $\mu$m in different vertical velocities are increasing from lower to higher altitudes till 7.5-10 km range. In the upper-levels, this separation is relatively smaller. Figure 7 also shows increasing number concentration of larger particles with the altitude, consistent with Yang et al. (2016a) and we can see this increasing trend extending from mixed phase regime (which other studies have focused on) to beyond the homogenous freezing level.

Simultaneously, a faster increase in number concentrations of larger particles ($D_p > 100$ $\mu$m) in strongest draft regions can be noted from lower to higher altitudes. Updrafts carry particles from below, where they are subject to different growth processes according to the thermodynamic conditions. Analysis of PSDs in mixed-phase clouds have shown that there are more millimeter sized particles in stronger updrafts at the cloud top, suggesting more larger ice particles generated in stronger



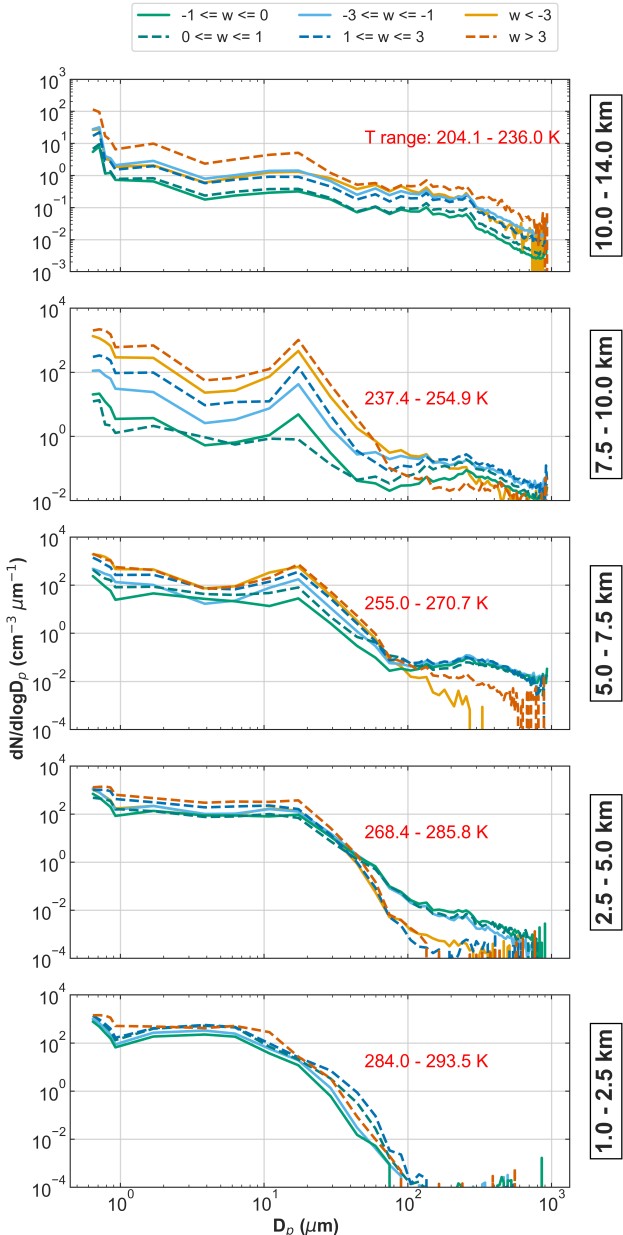

**Figure 7.** Particle size distribution of in-cloud up and downdrafts in different altitudes. The dashed lines represent updrafts and the solid lines represent downdrafts. The colours represent vertical velocity bins.

updrafts (Yang et al., 2016a). It is unclear what causes the presence of larger particles in downdrafts and its increasing number
concentrations with altitude and draft intensity.



The results presented in this section outlines the effect of vertical velocity on mean PSDs. Given the lack of directly comparable data, it is difficult to comprehensively evaluate these results with other studies. To enable an indirect analysis, we assume, if we have measurements of a developing system, this contains stronger drafts. Thus, our strong drafts could be compared with young deep convective clouds such as in Frey et al. (2011) (see Fig. 19). This could imply that the mean PSDs from young outflow regions can be compared with upper-level mean PSDs in the stronger vertical velocity bin shown in Fig. 7. PSDs show similar structure for both upper-level PSDs shown in this study and the PSD of young outflow of MCS shown in Frey et al. (2011). Additionally, identical to upper-level PSDs, young outflow clouds are observed to have the highest number concentration and the largest maximum particle size. Previous studies have shown that aerosol particle concentration is the primary driver for the vertical profiles of effective diameter and droplet concentration in the warm phase of Amazonian convective clouds, while updraft speeds have a modulating role in the latter and in total condensed water (e.g. Cecchini et al., 2017a). Background aerosol conditions affect the broadening of PSDs in the growing cumuli through various mechanisms such as secondary droplet activation (in polluted conditions) and collision-coalescence (in clean conditions) (Hernández Pardo et al., 2021) and strong updrafts just below to freezing level increase the condensational growth of droplets (Cecchini et al., 2017b). However, these effects primarily impact warm-phase cloud processes and the evolution of the mixed-phase as droplets cross the freezing level but do not affect the upper-levels, which are distant from the cloud base.

The downdraft size distributions (solid lines) have very similar distribution to that of updrafts (dashed lines) with similar absolute vertical velocities except for 5-7.5 km range. In this altitude range, the mean size distribution for strong downdrafts (w < -3 m s$^{-1}$) shows lower number concentrations for particles with $D_p$ > 100 $\mu$m than updraft. This could be a related to fewer measurements in this altitude range, making the results less representative especially for stronger drafts. A potential explanation for the similar distributions could be the existence of large eddies that allow mixing between updrafts and downdrafts plays a role in such similar distributions. For such mixing to occur, the updrafts and downdrafts should be located next to each other, so that turbulent eddies can turn between the drafts, enabling the mixing. This could lead to similar properties in up and downdrafts. Unfortunately, it is not possible to prove that with single penetration data from aircrafts. Additionally, in the upper levels, the cloud particles might have undergone some processing as they are part of the stratiform cloud deck and the formation of hydrometeors has mostly taken place in the growing cumuli. If the measurements are not part of updraft-downdraft structure (located next to each other), a different explanation is needed for the similarities in microphysical properties of updrafts and downdrafts. This alternative explanation could contain a very quick microphysical reaction time scale. In that case, not the history of the draft is important but the current conditions that play a major role for determining the microphysical properties.

## 5 Conclusions

The characteristics of in-cloud downdrafts in deep convective clouds are studied using the aircraft measurements from ACRIDICON-CHUVA campaign. A focus is given on upper-levels (10 - 14 km) due to a lack of knowledge on the upper level downdrafts, and potentially different mechanisms that drive/maintain downdrafts in ice clouds compared to water clouds. A caveat is that the aircraft did not pass through the most intense regions of convection. As a result, the findings presented here are not rep-



resentative of the convective core. However, the cloud deck covers up to 80 % of a mature convective system (Machado and

Rossow, 1993) and thus the measurements analysed here represent large part of the system. Major findings from the study are given as follows:

– Mean draft diameters increase with altitude and the mean air mass fluxes decrease with altitude in both updrafts and downdrafts.

– The CWC measurements show no obvious trend with altitude and vertical velocity of detected downdrafts and updrafts,
contradicting the effect of condensate loading on the downdraft strength in the clouds.

– In the upper-levels, high negative vertical velocities (w > 2 ms$^{-1}$) are found in supersaturated regime that contradict the view of downdrafts driven by cooling through evaporation/sublimation. The drivers and maintenance of such downdrafts will be explored using high resolution large eddy simulations in a follow up study.

– Cloud droplet number concentrations in downdraft and updraft regions are very similar. For stronger vertical velocities
($|w| > 1$ ms$^{-1}$), droplet number concentrations show more linear trend with vertical velocity.

– The mean particle size distribution shows an increase in maximum particle size with altitude, consistent with previous studies. It also shows relatively higher particle concentrations for particles $D_p < 100$ $\mu$m in strongest draft regions. The numbers of larger particles ($D_p > 100$ $\mu$m) show a faster increase with altitude in strongest draft regions.

– The number concentrations in downdrafts are comparable to that of updrafts of similar strength at the same altitude. This
could imply the existence of larger eddies that allow mixing between updrafts and downdrafts located next to each other.

Evidently, interactions of upper-level downdrafts with microphysics are not readily observed in terms of empirical relationships due to its complexity. However, analyses presented in this study were able to point out the effects of such interactions through size distributions of cloud particles and CWC. Regarding mean air mass flux and CWC, the downdraft properties are very similar to updraft properties. These results clearly point out the need for more research to understand how downdrafts are
driven/maintained (in upper, but maybe also lower levels), utilising more observations and numerical modelling in future.

*Data availability.* The data are available through mission-based databases. They can be accessed after signing a data agreement. The permanent URL for ACRIDICON-CHUVA via the DLR HALO database is: https://halo-db.pa.op.dlr.de/mission/, last access 3 February 2025.



## Appendix A:  Regression parameters from Fig. 6

The regression parameters for the updrafts and downdrafts shown in figure 6 is given in the Table A1 and Table A2 respectively.

**Table A1.** Regression parameters for updrafts in different $RH_{ice}$ regimes in Fig. 6

| RHi_regime | slope | intercept | $R^2$ |
|---|---|---|---|
| $RH_i < 90$, CWC<300 | -0.0289 | 0.2474 | 0.0033 |
| $RH_i < 90$, CWC>300 | -0.1128 | 0.9335 | 0.0013 |
| $90 > RH_i > 110$ | 0.3448 | 0.2258 | 0.0378 |
| $RH_i > 110$ | 0.3757 | 1.7049 | 0.1269 |

**Table A2.** Regression parameters for downdrafts in different $RH_{ice}$ regimes in Fig. 6

| RHi_regime | slope | intercept | $R^2$ |
|---|---|---|---|
| $RH_i < 90$, CWC<300 | -0.0024 | 0.1091 | 0.00008 |
| $RH_i < 90$, CWC>300 | 0.1597 | 1.4038 | 0.01703 |
| $90 > RH_i > 110$ | -0.6422 | 0.7885 | 0.05953 |
| $RH_i > 110$ | -1.8124 | 0.2301 | 0.2867 |



*Author contributions.*   SK: conceptualized the study, designed the methodology, performed the analysis and wrote the manuscript, WF: con-
ceptualized the study, contributed to the methodology and data interpretation, provided critical revisions and feedback, JVGdA: contributed
to the methodology and data interpretation, provided critical revisions and feedback, LATM: provided critical revisions and feedback, MK:
Instrument PI for NIXE-CAPS, provided critical revisions and feedback, assisted with data processing, AA: Instrument PI for NIXE-CAPS,
provided critical revisions and feedback, MZ: Instrument PI for BAHAMAS, provided critical revisions and feedback. All authors reviewed
and approved the final manuscript.

*Competing interests.*   At least one of the (co-)authors is a member of the editorial board of Atmospheric Chemistry and Physics.

*Acknowledgements.*   The authors would like to express their gratitude to ACRIDICON-CHUVA campaign team, including scientists, flight
crews, and technical staff, for their efforts in data collection and management.



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
