# Peer review of "Unexpected characteristics of convective downdrafts in the upper-levels of tropical deep convective clouds"

_EGUsphere, 2025_

## Referee Comment (RC2)

Reviewer of : „Unexpected characteristics of convective clouds downdrafts in the upper-levels of tropical deep convective clouds"

The authors use the aircraft measurements from the ACRIDICON-CHUVA field campaign to analyze the characteristics of updraft and downdraft in the cloud deck of convective systems. The characterization is related to shape, dynamics, and microphysical properties of drafts in the upper troposphere, but it also includes altitudes in the mid and lower troposphere. The results suggest that downdrafts and updrafts share similar properties. Drafts tend to increase their diameter with altitude, but the mass flux decreases with height. This is explained by the reduction in the vertical velocity. At upper levels (10-14 km), drafts with a diameter smaller than 1000 m are the most frequent, but their mass flux is lower than larger drafts, which are less common. The authors show no linear relationship between the amount of cloud water content and the velocity of drafts at different altitudes. This, in the authors' interpretation, suggests that the drag force due to more condensed water is not applicable for the formation of downdrafts. The authors argue that strong downdrafts are more common in the case of a supersaturated state (RH$_i$ > 110%), and not in a subsaturated state (RH$_i$ < 90%), which is expected because a subsaturated state would require more energy (sublimation) and, as a consequence, more cooling. Moreover, the more intense downdrafts in a supersaturated state are related to high values of droplet number concentration. Strong downdrafts between 5 and 10 km also tend to be related to high concentrations of particles with sizes less than 100 $\mu$m.

The analysis presented by the authors shows what they state, but I find the whole manuscript too simple in the analysis and interpretation of the results. It's difficult to get the idea the authors want to transmit. Do they want to say that the way we think about downdrafts is wrong? Or do the characteristics of downdrafts depend on the region or the sampling method? The dynamic of the manuscript is to present a result a then compare it to the other studies. If the comparison goes in the same direction, the authors state that the results are in accordance with the literature. When it is not the case, the authors give different hypotheses to explain the discrepancy, but there is no analysis supporting their hypotheses. This way of interpretation gives the manuscript a direction of too speculative. In the following lines, I give my major concerns about the manuscript:

1. The objective of the paper is difficult to the asses. The introduction, which in my opinion is unnecessarily long, does not provide the problem that the authors aim to tackle. The motivation presented by the authors is that there is not enough literature about observations of upper-level drafts. With this statement, I was expecting an overview of the different structures of downdrafts in convective systems, but the authors only focus on one campaign, and according to the manuscript, the object of sampling is restricted to cloud decks around convective systems, avoiding the convective core. This points out that the sampling region could also play a role in the results of the manuscript. I would suggest stating clearly in the introduction what is the scientific questions that the authors are tackling in the manuscript. Are they stating that not all the downdraft shows the same properties? I also suggest stating clearly what the limitations of their study are.

   Regarding the limitations of the study, I was wondering if the results would change if upper-level downdrafts inside convective cores were included.

2. The results section is difficult to read because it is arduous to get a continuous storyline. The authors explain their results with respect to other studies, looking only for similarities or differences, but there is a lack of deepening the analysis to explain the possible hypotheses stated by the authors. If the authors want to compare their results, I suggest opening a Discussion section.

3. In the same direction as point 2, the authors stated that neither the loading nor the evaporative cooling can explain the downdraft velocity. If this is the case, what is the hypothesis that the authors propose, and can they prove it? The manuscript will tremendously benefit from more analysis to prove or disprove those hypotheses.

4. I had difficulties understanding the mass flux discussion. The authors find that upper-level drafts have less mass flux than lower-level drafts. The discussion in the paper suggests that the vertical velocity is the one explaining the decrease in the mass flux with altitude. So, I was wondering if the authors expected that the mass flux of upper-level and lower-level drafts would be equal. If this is the case, please state this clearly in the manuscript. Moreover, assuming again that the mass flux should be conserved, are the changes of mass flux and velocity related to entrainment and detrainment?

   If it is not expected that the mass flux in upper-level and low-level drafts will be equal, what is the reason for the comparison?

5. The conclusion summarizes the main points of the manuscript, and in the actual structure and storyline of the manuscript, it gives a feel that the document is most like a collection of different analyses rather than addressing a scientific question. I would suggest addressing points 1 to 4 and changing the conclusion section depending on the outcome. Moreover, I find it hard to imagine large eddies communicating dropplets between downdrafts and updrafts without affecting their entropy.

Specific comments:

Line 11: What is Dp? And is it 100 $\mu$m?

Lines 11-12: Increases faster than what?

Lines 32-33: "However, the earlier observations …" I agree that this is part of the region analyzed in the manuscript, but as the authors stated, they only analyzed cloud decks.

Lines 101-103: Are you studying the whole spectrum of downdrafts or just a subset of this? What about drafts in convective cores?

Lines 176-178: How many samples do you have in the lower troposphere compared to the upper troposphere? How much does the PDF of the draft diameter vary when the same number of samples is chosen randomly in the lower and upper troposphere?

Lines 196-197: "Drafts in lower altitudes …" I see your point, but I was wondering whether the decrease in density with altitude also explains the difference in the mass flux between upper-level and lower-level drafts.

Lines 212-215: What is the point of comparing if this is a different method?

Lines 219-220: Does it mean that the other 70% come from wider drafts?

Lines 226-228: "We would like to ..." How do you think that the sampling method affected the results?

Figure 4: If I understood correctly, every point is a draft. So, what about dividing the cloud water content by the draft diameter?  This is to see if the concentration of cloud water content with respect to the diameter shows a relationship with vertical velocity.

Lines 236-237: I did not understand the logic of the two sentences. First, atmospheric motion is influenced by hydrometeors, but in the following sentence, which is an example, updrafts influence supersaturation. So, are hydrometeors affecting atmospheric motion or the other way around?

Lines 255-256: "We observe stronger downdrafts …" Are you sure? What I see is that the subsaturated state has equally strong downward vertical velocity as the supersaturated downdraft (< -1 m s-1).

Lines 273-275: How confident are you that this method removes the influence of mesoscale draft in the vertical velocity?

Line 302: "Larger particles", what does it mean larger particles? $D_p > 100$ $\mu$m , or are you talking about particles close to 100 $\mu$m.

---

## Author Comment (AC1)

Reply to Reviewer #1

Thank you for providing critical and constructive comments regarding our manuscript. Following your and the second reviewer's comments, we have revised the manuscript significantly. The introduction has been rewritten completely to provide a more focused and clearer storyline. The results and discussion sections have been separated to improve the clarity, providing the detailed characterisation in the results section, while presenting respective discussion with more elaborate explanations in the discussion section.

Below, we provide answers to each of the reviewer's comments, with the original comments in red, our clarifications and answers in black, and newly added text in blue. Any reference to lines in our answers is given with respect to the original manuscript.

**Major Comments:**

1. The paper claims that a lack of correlation between CWC and downdraft intensity contradicts the role of condensate loading. This conclusion is not justified. Vertical velocity is influenced by multiple, competing factors, including pressure perturbations, phase changes, mixing, etc, and may not exhibit straightforward relationships with single microphysical variables. Grant et al. (2022) is cited in the manuscript to justify this hypothesis; however, the cited paper refers to the relationship between vertical velocity and the rate of condensate production, not directly to CWC, and this relationship was only shown for updrafts. Adiabatic compression during descent decreases a parcel's supersaturation, eventually leading to partial or total evaporation or sublimation of the condensate. This feedback can cause CWC to decrease as vertical velocity becomes more negative, making the expectation of a simple positive correlation between CWC and |w| in downdrafts, as stated in the manuscript, at least questionable.

   We thank the reviewer for pointing this out.

   The idea to examine the CWC was to see its effect on modulating the dynamical and thermodynamic responses in updrafts and downdrafts. This was motivated by the discussions from past literature (e.g., Knupp and Cotton, 1985) on downdraft initiation and maintenance in the clouds. According to them, hydrometeor loading and latent cooling are fundamental to the formation and maintenance of the downdrafts. Since CWC is the amount of cloud water present in the air, it represents the magnitude of hydrometeor loading.

   To give a clear context for the comparison with Grant et al. (2022), we have added the following information to line 240

   *They show that the net effects of all microphysical processes, such as condensation, evaporation, deposition, sublimation, and cloud droplet activation, vary linearly with updraft velocity. However, information on the rate of condensate converted from water vapour is not available in the aircraft data used in this study, as information on the timely evolution (history) of the measured variables is not available.*

We have added an explanation of the interplay between CWC and vertical velocity in the downdrafts to line 243

*The adiabatic compression and warming in downdrafts can determine the amount of CWC. This reduces the relative humidity of the downdraft parcel and, when it eventually reaches sub-saturation, reduces CWC by evaporation/sublimation. Furthermore, stronger downdrafts may reduce CWC faster due to increased adiabatic compression and consequent warming.*

Our findings challenge the 'classical' view of downdraft initiation and maintenance, by being subsaturated and driven by condensate loading, indicating that this view may be incomplete. We elaborate more on the classical versus new hypotheses in the reply to reviewer #2, comment 3. Therefore, we believe that our results can be useful for evaluating model performance and interpreting model results. Furthermore, it can help inform the flight strategies of future airborne field campaigns.

To provide more clarity to our study, we have split up the results and discussion. This gives us a self-contained results section that provides a thorough analysis of the unique observations, including their statistics. The discussion section here would be more specific to incorporating plausible physical explanations.

2. The interpretation that negative vertical velocities in ice-supersaturated air masses contradict the effects of sublimation/evaporation is an oversimplification. Supersaturation at the time of observation does not preclude earlier evaporation/sublimation that may have initiated the downdraft. Vertical velocity at a point reflects accumulated forcing along a parcel's trajectory, not only the local, instantaneous forcing. Without trajectory or time-resolved data, causal conclusions about downdraft drivers are not warranted. The authors acknowledge the possibility that the downdraft was driven by evaporation or sublimation prior to the measurement (lines 263-264), but still arrive at the opposite conclusion.

We agree that the supersaturation at the time of measurement does not preclude the earlier sub-saturation of the parcel. However, it points out a scenario where the evaporation no longer maintains the downdrafts by latent cooling. Moreover, if a downdraft parcel is supersaturated at the time of measurement, we think it is highly unlikely to be subsaturated at a previous time. This is because, during its descent, an initially subsaturated parcel would increase its temperature, thus maintaining its subsaturated state. Evaporation or sublimation of cloud particles in such a scenario would, at most, lead to a saturated state of the air parcel, but not to a supersaturated state. Additionally, the adiabatic compression decreases the supersaturation of the downdraft parcel, as mentioned in the reviewer's first comment. An exception to this is if there is an additional influx of moisture, e.g. through mixing between a neighbouring supersaturated updraft and the downdraft.

We have replaced lines 258 – 263 in the manuscript with the following lines:

*Supersaturated regions of downdrafts are unlikely to be subsaturated at a previous time. If they were subsaturated before the time of measurement, this state would have been retained due to the increase in temperature and a subsequent decrease in relative humidity and accordingly, cloud water content within the downdraft as it descends to lower altitudes. An exception is if there is an additional moisture supply (e.g., through mixing) from neighbouring regions, such as supersaturated updrafts, which can bring the downdraft regions to a saturated or supersaturated state.*

Although there are studies using numerical models that have pointed out the existence of supersaturation even in downdrafts (D'Alessandro et al., 2017), a concise explanation is lacking. Furthermore, a closer evaluation of the supersaturated points shown in Figure 5 in the manuscript revealed that the supersaturation is part of several downdrafts. Thus, these features are not anecdotal and require more detailed research.

In the manuscript line 273, we have added

*The measurements show that the supersaturated points in Figure 5 originate from several downdrafts from different flights. Although there are studies using numerical models that have pointed out the existence of supersaturation in downdrafts in ice clouds (D'Alessandro, J. J., et al., 2017), a concise explanation is lacking.*

We provide a potential explanation for the existence of supersaturated downdrafts in the answer to the following point.

3. The suggestion that "large eddies" explain the similarity in particle size distributions between strong up- and downdrafts is vague and potentially inconsistent with the data. If the strongest drafts are most often relatively narrow, as indicated in the manuscript, then the mixing eddies that connect them should be small as well. In fact, larger eddies would typically imply longer times spent within either updrafts or downdrafts, thus enhancing, rather than reducing, differences in particle characteristics compared to smaller eddies with shorter updraft and downdraft segments. The manuscript would benefit from a clearer definition of eddy scale and a more explicit explanation of the proposed mixing mechanism.

Using an undefined length scale for the eddies here can be misleading. Therefore, we will use "eddies" (without any length scale) throughout the manuscript (lines 16, 330, 360), and add a more elaborate explanation in the discussion of these.

As the reviewer pointed out, the strongest drafts are narrower; hence, the eddy associated with them would also be smaller. However, quantifying the eddy scale from auto-correlation or similar methods is difficult due to the high aircraft speed ($\sim 150$ m/s) and the accordingly short time spent in smaller drafts.

In the manuscript we have added in section 4.3 (line 333):

*Due to the high aircraft speed (~150 m/s), resulting in short times spent in the smaller drafts, it is not possible to calculate the actual eddy scales using methods like auto-correlation, which require data for a significant period inside the drafts.*

The proposed mixing mechanism involves a simple transfer of cloud particles between updrafts and downdrafts, as this "sharing" could result in similar PSDs. To clarify further, we analysed the updraft-downdraft structures during different flights and their associated PSDs. Figure R1 shows two such cases and the average PSDs related to the updrafts, downdrafts, and the area outside the draft structure. This shows that the PSDs of updrafts and downdrafts are comparable, while one could have expected differences in the PSDs from the updrafts (coming from an altitude below) and downdrafts (coming from an altitude above), c.f. Figure 7 of the original manuscript which shows the change in size distribution with altitude. On the other hand, the PSD of regions outside the draft structure differs and has lower concentrations. This could be indicative of a direct link between the updraft and downdraft that are located next to each other, as they show similarity not only

[Figure]

Figure R1: Particle Size distribution and time series of w, altitude, and RH of the corresponding flight segments. Similarities in updraft (grey shading) and downdraft (red shading) PSDs are visible, while those from outside the draft structure (yellow shading on the right) exhibit lower number concentrations.

in the PSDs but also in e.g. relative humidity as shown here.

4. While the title and conclusions emphasize downdrafts, the figures and analyses give comparable attention to updrafts. The manuscript might be better positioned as a study of upper-level cloud properties, particularly of anvil regions, focusing on microphysical

structure and variability, rather than attempting to infer cloud dynamics from limited information.

While the original aim of the study was to elucidate the downdrafts, we agree, that in the manuscript as provided, the title might be misleading in that regard. Therefore, we chose to revise the title to: *Upper-level characteristics of updrafts and downdrafts in tropical deep convective clouds.*

We also paid attention to this comment while rewriting the introduction.

5. The manuscript would benefit from thorough professional proofreading. There are frequent issues with article usage, and sentence structure overall, as well as redundant phrasing, which at times reduce the clarity of the scientific argument.
Along with the significant revision, we have carefully checked and corrected the manuscript for grammatical errors and redundant phrasing.

**Minor Comments:**

The introduction is overly long and lacks a clear narrative structure. It moves back and forth between studies without establishing a coherent line of reasoning. In several places, relatively recent studies are cited to explain long-established mechanisms, which can be misleading. A more concise and focused literature review is recommended.
We completely rewrote the introduction in order to provide a clearer storyline, as also stated at the beginning of our reply.

Figures 1 and 2: It is unclear whether these show one-dimensional PDFs at each height or joint PDFs as a function of height and draft diameter/mass flux. If they are 1D slices, please show the number of observations per height bin.
The PDFs shown in Fig. 1 and Fig. 2 represent joint PDFs in height and draft diameter/mass flux. The density values are calculated based on the bin width of both variables in x and y axes. We have clarified this in the text and the figure caption in the revised manuscript, accordingly.

Confidence intervals or error bars should also be added to the mean and percentile curves in panels b.

We thank the reviewer for the suggestion. We have added the confidence interval for mean and percentile curves in panel b of Figures 1 and 2. Additionally, we have modified the number of drafts in Figure 1b to draft fraction, as suggested by the editor.

[Figure]

Figure 1 (revised) : Altitude-wise draft diameter statistics of all in-cloud drafts. (a) Joint Probability Density Function of Draft diameter and altitude (b) mean (blue) and 95th percentile (yellow) values of diameter of drafs. 90% confidence intervals for mean and percentile curves are indicated by the error bars.

[Figure]

Figure 2 (revised) : Altitude wise air mass flux statistics of all in-cloud drafts (a) Joint Probability Density Function of Air mass flux and altitude (b) mean (blue), 5th percentile (red) and 95th percentile (yellow) of air mass flux. 90% confidence intervals for mean and percentile curves are indicated by the error bars.

Figure 5: Consider using a heatmap or 2D histogram to show point density more clearly. The current scatterplots suffer from significant overlap of data points, making it difficult to interpret the underlying distribution.

We have revised the Fig. 5 from scatter plot to a 2D histogram to make it more comprehensive.

[Figure]

Figure 5 (revised) : Histogram of Cloud water content (ppmv) versus vertical velocity (m/s) for upper-levels (10-14 km) in regimes based on $RH_{ice}$. (a) Subsaturated ($RH_{ice} < 90\%$), (b) Transition/Intermediate ($90\% < RH_{ice} < 110\%$), and (c) Supersaturated ($RH_{ice} > 110\%$).

Figure 7: The green and blue shades are hard to distinguish, please choose more contrasting colors.

We have revised the colors in Fig. 7 to improve the readability also keeping in mind colour vision impairments.

[Figure]

6. how do varying sample sizes in each vertical velocity and height bin affect the calculated distributions? Are the results statistically robust across all w-z bins? Some quantification of uncertainty or sampling error would be helpful.

Fig. R2, shows the number of observations used in calculating PSD at different altitudes and vertical velocity bins. The number of observations for the strongest vertical motions is generally lower. This is expected, as it is difficult to sample the strongest parts of the convection with aircraft.

[Figure]

Figure R2: Number of observations in each vertical velocity bin at different altitudes.

**References**

D'Alessandro, J. J., Diao, M., Wu, C., Liu, X., Chen, M., Morrison, H., Eidhammer, T., Jensen, J. B., Bansemer, A., Zondlo, M. A., & DiGangi, J. P. (2017). Dynamical conditions of ice supersaturation and ice nucleation in convective systems: A comparative analysis between in situ aircraft observations and WRF simulations. *Journal of Geophysical Research: Atmospheres*, *122*(5), 2844–2866. https://doi.org/10.1002/2016JD025994

Knupp, K. R., & Cotton, W. R. (1985). Convective cloud downdraft structure: An interpretive survey. In *Reviews of Geophysics* (Vol. 23, Issue 2, pp. 183–215). https://doi.org/10.1029/RG023i002p00183

---

## Author Comment (AC2)

Reply to Reviewer #2

We thank the reviewer for providing comments on our manuscript. Following your and the first reviewer's comments, we have revised the manuscript significantly. The introduction has been rewritten completely to provide a more focused and clearer storyline. The results and discussion sections have been separated to improve clarity, providing detailed characterisation in the results section, while presenting respective discussion with more elaborate explanations in the discussion section. Below, we provide answers to the reviewer's comments, with the original comments in red, our clarifications and answers in black, and newly added text in blue. Any reference to lines in our answers is given with respect to the original manuscript (not the revised version).

[…general comment by the reviewer…] The authors argue that strong downdrafts are more common in the case of a supersaturated state (RHi > 110%), and not in a subsaturated state (RHi < 90%), which is expected because a subsaturated state would require more energy (sublimation) and, as a consequence, more cooling.

Moreover, the more intense downdrafts in a supersaturated state are related to high values of droplet number concentration. Strong downdrafts between 5 and 10 km also tend to be related to high concentrations of particles with sizes less than 100 um. The analysis presented by the authors shows what they state, but I find the whole manuscript too simple in the analysis and interpretation of the results. It is difficult to get the idea the authors want to transmit.

Do they want to say that the way we think about downdrafts is wrong?

Or do the characteristics of downdrafts depend on the region or the sampling method? The dynamic of the manuscript is to present a result a then compare it to the other studies. If the comparison goes in the same direction, the authors state that the results are in accordance with the literature. When it is not the case, the authors give different hypotheses to explain the discrepancy, but there is no analysis supporting their hypotheses. This way of interpretation gives the manuscript a direction of too speculative.

We thank the Anonymous reviewer #2 for the detailed remark on the manuscript as a whole. We have now restructured the manuscript in order to improve the clarity of the analysis and, at the same time, bring more substance and explanations to our discussion. We have split the results and discussion into two self-standing sections. This provides a self-contained results section that offers a thorough analysis of the unique observations, including their statistics. The discussion section is used to elaborate on similarities and differences we have found in comparison to previous studies.

Before addressing the specific comments, we would like to clarify that we want to point out the existence of supersaturation in downdrafts. However, we do not want to argue that the downdrafts are more common based on the state they are in (subsaturated vs supersaturated).

**Major comments:**

In the following lines, I give my major concerns about the manuscript:

1. The objective of the paper is difficult to the asses. The introduction, which in my opinion is unnecessarily long, does not provide the problem that the authors aim to tackle. The motivation presented by the authors is that there is not enough literature about observations of upper-level drafts. […] the object of sampling is restricted to cloud decks around convective systems, avoiding the convective core. This points out that the sampling region could also play a role in the results of the manuscript. I would suggest stating clearly in the introduction what is the scientific questions that the authors are tackling in the manuscript […]

Thank you for pointing out the difficulty in assessing the objective of this study. The objective of this study is to gain a deeper understanding of the characteristics of deep convective cloud anvils, which cover large areas and thus constitute a significant part of the convective system. To achieve this, we use rare in situ aircraft observations from high altitudes (up to 14 km) during the ACRIDICON-CHUVA campaign over the Amazon. In particular, we provide details on the dynamical, thermodynamic, and microphysical properties of updrafts and downdrafts, which will help to inform numerical model simulations. As previous studies (including modelling) mostly focus on the core region of deep convection, more knowledge about the properties in the anvil parts of the clouds is needed. We will include these objectives clearly in the introduction to enhance the clarity.

[…]Are they stating that not all the downdraft shows the same properties? I also suggest stating clearly what the limitations of their study are. Regarding the limitations of the study, I was wondering if the results would change if upper-level downdrafts inside convective cores were included.

The inhomogeneities in downdraft characteristics are one of the takeaways of our study. For example, some downdrafts appear in supersaturated regions, and others can be completely subsaturated. Particle size distributions in updrafts and downdrafts show similar characteristics in upper levels. These findings help us to enhance our understanding of draft characteristics and can be used to evaluate model performance.

Certainly, the sampling of different parts of clouds can affect the observed characteristics. For example, the convective cores have stronger updrafts, resulting in higher updraft and downdraft mass fluxes (Wang et al., 2020). Additionally, stronger updrafts can carry larger hydrometeors aloft, and thus, they can affect the particle size distribution.

2. The results section is difficult to read because it is arduous to get a continuous storyline. The authors explain their results with respect to other studies, looking only for similarities or differences, but there is a lack of deepening the analysis to explain the possible hypotheses stated by the authors. If the authors want to compare their results, I suggest opening a Discussion section.

We thank the reviewer for the suggestion. While we previously thought it might be easier to keep descriptive results of particular aspects and the respective discussion together, we have now come to the conclusion that it is indeed better to separate results and discussion completely, and split the sections up accordingly.

3. In the same direction as point 2, the authors stated that neither the loading nor the evaporative cooling can explain the downdraft velocity. If this is the case, what is the hypothesis that the authors propose, and can they prove it? The manuscript will tremendously benefit from more analysis to prove or disprove those hypotheses.

Indeed, the vertical velocity depends on several factors that can act simultaneously, such as the amount and the size/weight of hydrometeors, relative humidity, temperature, etc. Previous studies (Jorgensen and LeMone, 1989; Kamburova and Ludlam, 1966; Knupp and Cotton, 1985; Lucas et al., 1994) have identified condensate loading, related to the drag force of the hydrometeors, and evaporative cooling, relying on the downdrafts being subsaturated, as driving factors of the downdrafts. Our first hypothesis was that these drivers would also act in the upper cloud parts. However, our data shows that a significant amount of downdrafts (250 downdrafts) are in fact saturated or even supersaturated with respect to ice; thus, evaporative cooling (or, in ice clouds, sublimative cooling) cannot be the force driving or maintaining the downdrafts. To prove or disprove the condensate loading hypothesis, we looked at the relation between cloud water content, as a representative of condensate loading, versus vertical wind. Also, here, we cannot find a solid relation. This could be due to the fact that in ice clouds, particularly the larger particles are typically not spherical but have very complex shapes, which act very differently than spherical particles in terms of fall velocity. Thus, the effect of the drag force on the particles is significantly different. As the hypothesised drivers were identified in studies looking at warm clouds, we can say that our data indicates that these drivers are not playing (a major) role in ice cloud anvils. To understand what might drive/maintain the downdrafts in the deep convective cloud anvils, we use the schematic drawn by Houze et al. (1989), which in the original version only shows only one older cell, but more older cells follow successively behind. We add these in the original figure below (as Houze et al. state, even though the older cell is weakening, it is characterised by an updraft core, followed by a convective-scale downdraft, depicted here by red arrows). While the older cells seem to be maintained by "inertia", we postulate that the downdrafts between the old updraft cores are basically maintained by the existence of the older updrafts (air goes up, must come down), thus, in a way, also maintained by inertia. We include this discussion in the revised

manuscript. With our airborne observations that consist of point measurements, we can, unfortunately, not prove this. However, we would like to employ a Large Eddy Simulation model (MicroHH) in a future study to look more closely at the structures of the anvil up- and downdrafts, investigating the "classical" driving forces (condensate loading and evaporative/sublimative cooling), the role of the older updrafts, and also the saturation sate of the downdrafts.

[Figure]

Adapted from Houze et al. (1989): Conceptual model of a squall line with a trailing stratiform area viewed in a vertical cross section oriented perpendicular to the convective line (i.e., parallel to its motion). Older convective cores are indicated by black ellipses and the associated downdrafts in red arrows.

4. I had difficulties understanding the mass flux discussion. The authors find that upper-level drafts have less mass flux than lower-level drafts. The discussion in the paper suggests that the vertical velocity is the one explaining the decrease in the mass flux with altitude. So, I was wondering if the authors expected that the mass flux of upper-level and lower-level drafts would be equal. If this is the case, please state this clearly in the manuscript.

Mass flux in clouds depends on air density, draft width, and vertical velocity. The vertical profile of draft width shows an increasing trend with altitude (Figure 1b) which would lead to an increase in mass flux. However, the mean mass flux decreases with altitude (Figure 2b), despite the linear relationship between mass flux and draft width (Figure 3). The reason for this is the decrease in mean vertical velocity and air density with altitude.

To clarify, the mass flux profile presented in this study is not entirely dependent on vertical velocity variations. It is also not expected to have similar mass flux values in higher altitudes and lower altitudes as the measured vertical velocities differ at different altitudes.

In order to make it clear, we have added in the line 195 :

*The majority of the drafts at higher altitudes show very weak mean vertical velocity and low air density. So, despite the fact that they are wider, the overall mass flux values are*

*lower. Drafts in lower altitudes correspond to less width but relatively high mean vertical velocity values and higher air density, resulting in higher mass flux.*

Moreover, assuming again that the mass flux should be conserved, are the changes of mass flux and velocity related to entrainment and detrainment? If it is not expected that the mass flux in upper-level and low-level drafts will be equal, what is the reason for the comparison?

We have not come across studies that specifically discuss mass flux conservation using aircraft data. Could you provide more clarification on the assumption of mass flux conservation and its context?

The comparison of the upper and lower levels aims to highlight the effect on average width and intensity of drafts on mass flux values at different altitudes. As previous studies have primarily focused on lower levels, we complement their work by extending our analysis to the upper levels. A key objective here is to establish the characteristics of the upper-level drafts, which remain unknown to date.

To make it clear, we have amended the following in the line 199:

*This discussion shows that upper-level drafts are wider in anvil clouds compared to the lower-level drafts observed during the campaign. It also provides an overview of the differences in draft characteristics between higher and lower altitudes, noting that previous studies mainly focused on lower altitudes.*

5. The conclusion summarizes the main points of the manuscript, and in the actual structure and storyline of the manuscript, it gives a feel that the document is most like a collection of different analyses rather than addressing a scientific question. I would suggest addressing points 1 to 4 and changing the conclusion section depending on the outcome. Moreover, I find it hard to imagine large eddies communicating dropplets between downdrafts and updrafts without affecting their entropy.

We have revised the conclusion section to enhance clarity and align it with the scientific question. To avoid the ambiguity regarding the eddy sizes, we have removed the word "large" from the manuscript. Additionally, regarding the comments made by reviewer #1, we clarified the concept of direct mixing between updrafts and downdrafts. The proximity of updrafts and downdrafts is shown to have some effect on properties such as PSDs and $RH_{ice}$. Please refer to the figure below, which shows two instances where the PSDs in the updraft-downdraft structure are remarkably similar and exhibit higher number concentration than the region which is not part of the draft. For more detailed information, we would like to refer to our answer to Comment 3 of Reviewer #1.

[Figure]

Figure R1: Particle Size distribution and time series of w, altitude, and RH of the corresponding flight segments. Similarities in updraft (grey shading) and downdraft (red shading) PSDs are visible, while those from outside the draft structure (yellow shading on the right) exhibit lower number concentrations.

Specific comments:

Line 11: What is Dp? And is it 100 um?

Dp represents the diameter of the cloud particle with the unit in micrometres (corrected in the revised manuscript).

Lines 11-12: Increases faster than what?

Faster than the weaker drafts. We have rephrased the sentence to

*Furthermore, the number concentration of larger particles (Dp > 100 um) increases faster in stronger drafts than that in weaker drafts as altitude increases.*

Lines 32-33: "However, the earlier observations…" I agree that this is part of the region analyzed in the manuscript, but as the authors stated, they only analyzed cloud decks.

The introduction has been rewritten completely, and therefore, this statement has been revised.

Lines 101-103: Are you studying the whole spectrum of downdrafts or just a subset of this? What about drafts in convective cores?

The study focuses on ACRIDICON-CHUVA campaign data and downdrafts encountered within. Major part of the data is sampled from higher altitudes which this study utilizes to study the draft characteristics. Since convective cores are not sampled in this campaign, it is not discussed in this study.

In the revised version of the manuscript, specifically in the introduction and discussion, we make this more clear.

Lines 176-178: How many samples do you have in the lower troposphere compared to the upper troposphere?

There is a total of 19722 data points after filtering. In which 16108 data points are from above 10km, and 3614 data points are from below 10km. In terms of drafts, 1478 drafts are from above 10 km, and 557 drafts are from below 10 km.

We include this information in the results section of the revised manuscript.

How much does the PDF of the draft diameter vary when the same number of samples is chosen randomly in the lower and upper troposphere?

We agree that there could be concerns regarding the uncertainty arising from the sample sizes. We have now included the 90% confidence interval estimate for mean and percentile curves in panel b of Figure 1 and 2. Additionally, we have modified the number of drafts in Figure 1b to draft fraction, as suggested by the editor. In a general sense, the uncertainty in the mean is smaller than the percentile curve. We provide this information in the revised manuscript.

Lines 196-197: "Drafts in lower altitudes…" I see your point, but I was wondering whether the decrease in density with altitude also explains the difference in the mass flux between upper-level and lower-level drafts.

We agree with the reviewer that the vertical profile of density would definitely affect the calculations of mass flux values. Please see our answer to comment 4.

[Figure]

Figure 1 (revised) : Altitude-wise draft diameter statistics of all in-cloud drafts. (a) Joint Probability Density Function of Draft diameter and altitude (b) mean (blue) and 95th percentile (yellow) values of diameter of drafs. 90 Confidence intervals for mean and percentile curves are indicated by the error bars.

[Figure]

Figure 2 (revised) : Altitude wise air mass flux statistics of all in-cloud drafts (a) Joint Probability Density Function of Air mass flux and altitude (b) mean (blue), 5th percentile (red) and 95th percentile (yellow) of air mass flux. Confidence intervals for mean and percentile curves are indicated by the error bars.

Lines 212-215: What is the point of comparing if this is a different method?

Thank you for the suggestion. We remove these lines in the revised manuscript.

Lines 219-220: Does it mean that the other 70% come from wider drafts?

Yes, the other 70% mass flux is contributed by drafts wider than 500m.

We added:

*Thus, the main contribution to up- and downdraft mass flux stems from the wider drafts (width > 500 m).*

Lines 226-228: "We would like to ..." How do you think that the sampling method affected the results?

Sampling at different locations of cloud (e.g., convective core) would affect the statistics and particle size distributions. Please see the answer to comment 1 for more detail.

Figure 4: If I understood correctly, every point is a draft. So, what about dividing the cloud water content by the draft diameter? This is to see if the concentration of cloud water content with respect to the diameter shows a relationship with vertical velocity.

Thank you for this suggestion. Unfortunately, we could not observe a conclusive relationship between CWC divided by draft width and vertical velocity (Figure R3). We would also like to clarify that each point in Figure 4 is an individual measurement, as calculating a representative value for each draft could lead to ignoring the fluctuations. This way, we can observe the actual value of the CWC.

[Figure]

Figure R3: (left) Vertical velocity vs Cloud Water Content (CWC) / draft width in updrafts (blue) and downdrafts (orange) points. (right) Joint Probability Density Function of vertical velocity and CWC/draft width.

Lines 236-237: I did not understand the logic of the two sentences. First, atmospheric motion is influenced by hydrometeors, but in the following sentence, which is an example, updrafts influence supersaturation. So, are hydrometeors affecting atmospheric motion or the other way around?

We thank the reviewer for pointing out the ambiguity in the sentence. We remove the first part of the sentence and start with

*The updrafts influence the supersaturation, thus the condensation process....*

Lines 255-256: "We observe stronger downdrafts…" Are you sure? What I see is that the subsaturated state has equally strong downward vertical velocity as the supersaturated downdraft (< -1 m s-1).

The sentence in the manuscript reads "We observe strong downdrafts in the supersaturated regime where, due to the supersaturation, sublimation is not possible."(not "stronger downdrafts"). This indicates the measurements with w < -1 m/s.

Lines 273-275: How confident are you that this method removes the influence of mesoscale draft in the vertical velocity?

This method is only to examine the regions in drafts where |w| > 1 m/s and not eliminate the mesoscale drafts. Due to the large length scale, these drafts have prolonged regions of very weak vertical velocities (|w| < 1 m/s) and large fluctuations in variables which introduce uncertainty, along with regions of significant vertical velocity (|w| > 1 m/s). Elimination of weaker vertical velocity points enables us to focus on the more active part of the drafts.

In line 275, we rephrased "higher intensity drafts" to "*more active parts of the drafts*".

Line 302: "Larger particles…", what does it mean larger particles? Dp >100 um , or are you talking about particles close to 100 um.

The term "larger particles" represent the particles with diameter (Dp) > 100 um. We added this information to improve clarity.

**References**

Houze, R. A., Rutledge, S. A., Biggerstaff, M. I., & Smull, B. F. (1989). Interpretation of Doppler Weather Radar Displays of Midlatitude Mesoscale Convective Systems. *Bulletin of the American Meteorological Society*, *70*(6), 608–619. https://doi.org/10.1175/1520-0477(1989)070<0608:IODWRD>2.0.CO;2

Jorgensen, D. P., & LeMone, M. A. (1989). Vertical Velocity Characteristics of Oceanic Convection. *Journal of Atmospheric Sciences*, *46*(5), 621–640. https://doi.org/10.1175/1520-0469(1989)046<0621:VVCOOC>2.0.CO;2

Kamburova, P. L., & Ludlam, F. H. (1966). Rainfall evaporation in thunderstorm downdraughts. *Quarterly Journal of the Royal Meteorological Society*, *92*(394), 510–518. https://doi.org/10.1002/qj.49709239407

Knupp, K. R., & Cotton, W. R. (1985). Convective cloud downdraft structure: An interpretive survey. In *Reviews of Geophysics* (Vol. 23, Issue 2, pp. 183–215). https://doi.org/10.1029/RG023i002p00183

Lucas, C., Zipser, E. J., & Lemone, M. A. (1994). Vertical Velocity in Oceanic Convection off Tropical Australia. *Journal of Atmospheric Sciences*, *51*(21), 3183–3193. https://doi.org/10.1175/1520-0469(1994)051<3183:VVIOCO>2.0.CO;2

Wang, D., Giangrande, S. E., Feng, Z., Hardin, J. C., & Prein, A. F. (2020). Updraft and Downdraft Core Size and Intensity as Revealed by Radar Wind Profilers: MCS Observations and Idealized Model Comparisons. *Journal of Geophysical Research: Atmospheres*, *125*(11). https://doi.org/10.1029/2019JD031774

---

## Author Comment (AC3)

Thank you very much for your thoughtful comments and feedback regarding our manuscript. We appreciate the time and effort you and the reviewers have invested in evaluating our work.

We acknowledge the concerns raised and recognise the need for substantial revisions. We believe that our study offers valuable insights for the community, and we would like to address the reviewers' and your comments in detail to strengthen the manuscript. To point out our aims more clearly: our analysis attempts to characterise the updrafts and downdrafts in the anvil of deep convective clouds, with their dynamic, thermodynamic, and microphysical properties, using aircraft observations (up to 14 km). The novel aspects are to study the anvil characteristics of deep convective clouds and whether we can explain their motion with concepts of hydrometeor loading and latent cooling. The difference in microphysical structure between clouds at lower altitudes and higher altitudes potentially contributes to their dynamical and thermodynamic properties, which are investigated in this study.

Below, we have provided specific responses to the points you raised. Additionally, we have addressed each of the reviewers' comments in the interactive discussion.

- How generalisable are the results obtained? Are they applicable to other convective situations?

  The campaign measured convective clouds at different life stages. As far as the upper levels (10-14 km) are concerned, the generalisation is robust. since the measurements are from the anvil of deep convective clouds (Wendisch et al., 2016). Amazonian deep convective cloud anvils typically occur at an altitude of 12 km, with anvil top heights reaching their maximum at 13.5 km (Dodson et al., 2018), thus our measurements, which were aimed at deep convective clouds but avoiding the core region (due to flight safety), are well withing the typical range of deep convective cloud anvils. Furthermore, the results from our study exhibit characteristics consistent with those reported in previous investigations. For example, at the microphysical level, the particle size distributions from higher altitudes resemble those from Frey et al. (2011) in which the particle size distribution of the anvil of the Mesoscale Convective System (MCS) over West Africa has been sampled. The draft characteristics, such as width and mass flux of drafts, show a similar vertical profile to those reported by Yang et al. (2016) for lower altitudes of convective clouds sampled over the tropics and midlatitudes. The novelty of our investigation is that we can extend these characteristics further to higher altitudes (>10 km). Since the in-situ observations of tropical deep convection at high altitudes are scarce, the present dataset is a valuable contribution to this limited body of knowledge. The main objective of this study is to gain a deeper understanding of the characteristics of deep convective cloud anvils, which cover large areas and thus constitute a significant part of the convective system. In the revised manuscript, we have explained and discussed the our findings within the context of the ACRIDICON-CHUVA campaign.

- I think it would be better to show updraft and downdraft fractions rather than number, because the number can more easily be biased by e.g. flying very high.

Thank you for the suggestion. In the revised version, the number of drafts in Figure 1b is now replaced with fractions (number of drafts in one altitude bin / total number of drafts) of updrafts and downdrafts. According to a comment by reviewer #2, we have given detailed information on number of drafts. In terms of drafts, 1478 drafts are from above 10 km, and 557 drafts are from below 10 km. Additionally, following the comment of reviewers' comments, we have added 90% confidence interval for mean and percentile curves.

[Figure]

Figure 1 (revised) : Altitude-wise draft diameter statistics of all in-cloud drafts. (a) Joint Probability Density Function of Draft diameter and altitude (b) mean (blue) and 95th percentile (yellow) values of diameter of drafts. 90 Confidence intervals for mean and percentile curves are indicated by the error bars.

- Reference to the simple 1D downdraft model of Srivastava (1985, https://doi.org/10.1175/1520-0469(1985)042<1004:ASMOED>2.0.CO;2) should be made. They e.g. show that downdrafts are more intense in higher relative humidity environments, because this makes dry downdrafts more negatively buoyant w.r.t. the environment.

Thank you for suggesting the inclusion of findings from Srivastava (1985). Environmental dependence is indeed a factor that influences draft characteristics. Srivastava (1985) provides insights to understand the downdrafts that occur in the sub-cloud layer, driven by

the evaporation of raindrops, and sensitive to the surrounding environmental conditions. However, our analysis does not show a direct relationship with latent cooling and cloud dynamics at higher altitudes. Moreover, we focus the environmental sensitivity is not the focus of the current study, thereby making it difficult to compare with the results of Srivastava (1985).

- The analyses appear rather random, and little effort seems to be made to synthesise results and provide perspectives.

Thank you for bringing attention to the concern regarding the study outline. In attempt to answer our research question, which is to explain the dynamical, thermodynamic and microphysical characteristics of anvil clouds with high-altitude in-situ observations, we analysed variables which relate to these aspects in the clouds. The results are essential to characterise the anvil cloud dynamics, which is complex due to the interplay of multiple processes occurring at various scales such as latent cooling/warming, entrainment/detrainment, ice microphysical interactions etc. We synthesised the results by providing a general overview (statistics of draft width, mass flux, etc.) and explaining specific interactions (hydrometeor loading and latent cooling). We found that the anvil cloud deck anvil downdrafts are not dominated by a single physical process. While the thermodynamic process does not explain downdrafts on its own, other processes such as radiative cooling and gravity waves produced by updrafts could contribute to them. However, it is not possible to evaluate the different hypotheses using only observational data. A future study will employ LES to evaluate the different contributions.

Our results carry relevance as they can also be used to inform numerical simulations. For instance, we show the draft width and airmass flux characteristics in upper levels, and that we can expect ice supersaturation in downdrafts of the anvils of the deep convective clouds. To provide more clarity, we rewrote the introduction and reorganised the results and discussion to mitigate ambiguity and provide perspectives. In the revised version, we pay special attention to clarify which observational findings require high-resolution modelling (cloud and turbulent resolved models) to refute them.

We believe that with our changes, we have significantly improved the manuscript and hope to have addressed all raised concerns appropriately.

Thank you again for your guidance and consideration.

**References**

- Dodson, J. B., Taylor, P. C., & Branson, M. (2018). Microphysical variability of Amazonian deep convective cores observed by CloudSat and simulated by a multi-scale modeling framework. *Atmospheric Chemistry and Physics*, *18*(9), 6493–6510. https://doi.org/10.5194/acp-18-6493-2018
- Frey, W., Borrmann, S., Kunkel, D., Weigel, R., De Reus, M., Schlager, H., Roiger, A., Voigt, C., Hoor, P., Curtius, J., Krämer, M., Schiller, C., Volk, C. M., Homan, C. D., Fierli, F., Di Donfrancesco, G., Ulanovsky, A., Ravegnani, F., Sitnikov, N. M., … Cairo, F. (2011). In situ measurements of tropical cloud properties in the West African Monsoon: Upper tropospheric ice clouds, mesoscale convective system outflow, and subvisual cirrus. *Atmospheric Chemistry and Physics*, *11*(12), 5569–5590. https://doi.org/10.5194/acp-11-5569-2011
- Wendisch, M., Poschl, U., Andreae, M. O., MacHado, L. A. T., Albrecht, R., Schlager, H., Rosenfeld, D., Martin, S. T., Abdelmonem, A., Afchine, A., Araujo, A. C., Artaxo, P., Aufmhoff, H., Barbosa, H. M. J., Borrmann, S., Braga, R., Buchholz, B., Cecchini, M. A., Costa, A., … Zoger, M. (2016). Acridicon-chuva campaign: Studying tropical deep convective clouds and precipitation over amazonia using the New German research aircraft HALO. *Bulletin of the American Meteorological Society*, *97*(10), 1885–1908. https://doi.org/10.1175/BAMS-D-14-00255.1
- Yang, J., Wang, Z., Heymsfield, A. J., & French, J. R. (2016). Characteristics of vertical air motion in isolated convective clouds. *Atmospheric Chemistry and Physics*, *16*(15), 10159–10173. https://doi.org/10.5194/acp-16-10159-2016